# An In Vitro Small Intestine Model Incorporating a Food Matrix and Bacterial Mock Community for Intestinal Function Testing

**DOI:** 10.3390/microorganisms11061419

**Published:** 2023-05-27

**Authors:** Mridu Malik, Jacob V. Tanzman, Sanat Kumar Dash, Cláudia N. H. Marques, Gretchen J. Mahler

**Affiliations:** 1Department of Biomedical Engineering, Binghamton University, Binghamton, NY 13902, USA; 2Binghamton Biofilm Research Center, Binghamton University, Binghamton, NY 13902, USA; 3Department of Biological Sciences, Binghamton University, Binghamton, NY 13902, USA

**Keywords:** intestinal epithelium, gut microbiome, diet, titanium dioxide nanoparticles, nutrient transport, microbial community

## Abstract

Consumed food travels through the gastrointestinal tract to reach the small intestine, where it interacts with the microbiota, forming a complex relationship with the dietary components. Here we present a complex in vitro cell culture model of the small intestine that includes human cells, digestion, a simulated meal, and a microbiota represented by a bacterial community consisting of *E. coli*, *L. rhamnosus*, *S. salivarius*, *B. bifidum,* and *E. faecalis*. This model was used to determine the effects of food-grade titanium dioxide nanoparticles (TiO_2_ NPs), a common food additive, on epithelial permeability, intestinal alkaline phosphatase activity, and nutrient transport across the epithelium. Physiologically relevant concentrations of TiO_2_ had no effect on intestinal permeability but caused an increase in triglyceride transport as part of the food model, which was reversed in the presence of bacteria. Individual bacterial species had no effect on glucose transport, but the bacterial community increased glucose transport, suggesting a change in bacterial behavior when in a community. Bacterial entrapment within the mucus layer was reduced with TiO_2_ exposure, which may be due to decreased mucus layer thickness. The combination of human cells, a synthetic meal, and a bacterial mock community provides an opportunity to understand the implications of nutritional changes on small intestinal function, including the microbiota.

## 1. Introduction

The human small intestine plays a significant role in nutrient absorption and homeostasis maintenance via host–microbe interactions. Due to the difficulty of accessing the human small intestine for monitoring dietary effects on gut homeostasis and nutrient absorption, in vitro models can be used to gain this knowledge. The intestinal environment is rich in bacterial populations ranging from 10^1^–10^3^ bacteria per gram of contents in the duodenum to 10^4^–10^7^ bacteria per gram of contents in the jejunum and ileum and 10^11^–10^12^ bacteria per gram of contents in the colon [1]. The environment of the small intestine is primarily composed of Gram-positive bacterial species such as *Lactocaseibacillus* and *Enterococcus* in the jejunum, *Streptococcus* in the duodenum and jejunum, and *Bifidum* bacilli [2] and Gram-negative *Escherichia coli* in the distal ileum [3]. The microbiota resides in the body as a community, competing for nutrients and metabolites, and are present in the mucus layer of the intestinal barrier facing the lumen. Under stressful conditions, disruption in the intestinal barrier and microbiota (dysbiosis) causes an altered permeability and disturbed intestinal function [4,5]. Intestinal alkaline phosphatase (IAP), a brush border membrane enzyme, which interacts with the microbiome to regulate the intestinal epithelial barrier, and its activity, together with intestinal permeability are often used as indicators of the integrity of the intestinal barrier.

The human small intestine, responsible for nutrient absorption and transport, is equipped with multiple transport proteins for nutrients such as glucose, proteins, and fatty acids [6]. In a healthy individual, 85% of carbohydrates, 66–95% of proteins, and all fats are absorbed in the small intestine before moving to the large intestine [7]. The gut microbiota draws its nutrition from the dietary components, and a change in diet can lead to changes in the microbial composition. For example, a change from a low-fat to a high-fat diet in mice led to a shift in their microbiota structure [8].

Nanoparticles (NPs) are used in the food industry to enhance the taste, aesthetics, or shelf life of food [9,10]. Food-grade titanium dioxide (TiO_2_), referred to as E171 in the European Union (EU), contains about 36% of particles in the nanoscale [11] and is widely used in the food industry to improve the color of foods such as candies, chewing gums, pastries, and sauces [10,11,12]. Regular consumption of TiO_2_-coated dietary products can be significant. Recently, the European Union issued a ban on the use of E171 food-grade TiO_2_ in Europe due to genotoxicity concerns [13], which is expected to impact the trade of the commodity, but in the absence of potential alternatives to the additive, its use is currently inevitable. As part of the dietary intake, these nanoparticles pass through the human gastrointestinal (GI) tract to enter the intestinal environment. TiO_2_ NPs can exhibit toxic effects through alteration of the gut microbiota and interference with nutrient transporters [11]. Some studies using TiO_2_ NPs show its involvement in increased inflammatory cytokine production, disruption of the structure of the intestinal epithelium and mucus layer, and gut homeostasis both in vitro [14] and in vivo [15,16]. Whereas other studies using rats show little or no effect of TiO_2_ ingestion but the elimination of it through the excretory system [17]. Such contradictory results call for the need to establish a standard study design to get more comparable and easily reproducible results.

This study uses a semi-synthetic food matrix supplemented with E171 food-grade TiO_2_ NPs to form a physiologically relevant food model for assessing the role of NPs on GI function. An in vitro model of the intestinal barrier was created using Caco-2 and HT29-MTX-E12 cell lines. The small intestinal microbiome is represented by adding planktonic cultures of individuals or a mix of bacterial species (forming a bacterial mock community), including Gram-negative *Escherichia coli*, and Gram-positive *Lacticaseibacillus rhamnosus, Streptococcus salivarius*, *Bifidobacterium bifidum,* and *Enterococcus faecalis*. The intestinal monolayer with bacteria was exposed to the food matrix, with or without NP, following an in vitro digestion to assess the effects of NPs and bacterial species on intestinal permeability, IAP activity, and nutrient transport. This study presents a novel in vitro intestinal model with a multispecies community of bacteria representing the microbiome, and a standard food matrix to mimic the small intestinal environment that can be used to assess how food additives affect the GI environment.

## 2. Materials and Methods

### 2.1. Intestinal Co-Culture

The human colon carcinoma Caco-2 cell line was purchased from the American Type Culture Collection (Manassas, VA, USA) at passage 17 and used for experimentation at passage 45–55. The HT29-MTX-E12 cell line was purchased from Sigma-Aldrich, St. Louis, MO, USA, and used in experiments at passage 50–60. These cells were grown in Dulbecco’s Modified Eagle Medium (DMEM, Gibco^®^ Thermo Fisher Scientific, Waltham, MA, USA), containing 4.5 g/L glucose and 10% (*v*/*v*) heat-inactivated fetal bovine serum (HI-FBS, Gibco^®^ Thermo Fisher Scientific, Waltham, MA, USA). The cells were maintained in 5% CO_2_ at 37 °C, and the media was changed every 48 h. Once the cells reached 80% confluency, they were passaged and seeded onto polycarbonate, 0.4 µm pore size, 0.33 cm^2^ membrane, 24-well Transwell^®^ inserts for permeability, nutrient transport, and bacterial viability studies, or onto polycarbonate 96-well culture plates for brush border membrane enzyme activity assessment. Prior to seeding, the wells/inserts were coated with rat tail Type I collagen (BD Biosciences, San Jose, CA, USA) at a concentration of 8 μg/cm^2^ for 1 h at room temperature. Both types of cells were stained with trypan blue, counted using a hemocytometer, and seeded at a density of 10^5^ cells/cm^2^ and a ratio of 75:25 (Caco-2:HT29). The co-culture of cells was grown for 14 days before experiments were conducted.

### 2.2. Bacterial Cell Culture

*Escherichia coli* (ATCC 53103), *Lacticaseibacillus rhamnosus* (ATCC 11775), *Streptococcus salivarius* SS2, *Bifidobacterium bifidum* VPI 1124, and *Enterococcus faecalis* NCTC 775 were purchased from the American Type Culture Collection. The bacterial species were cultured in brain heart infusion (BHI) broth medium supplemented with 0.5% glucose, 0.05% cysteine, and 0.1% agar. *E. coli* and *L. rhamnosus* cultures were grown at 37 °C, 5% CO_2_ for 24 and 48 h, respectively. *B. bifidum* was cultured in an anaerobic GasPak at 37 °C for 48 h. *S. salivarius* cultures were incubated under normal atmospheric conditions at 37 °C for 24 h. *E. faecalis* was cultured for 12 h at 37 °C, 5% CO_2_ condition. The bacterial stocks were diluted in 0.9% saline solution, plated on BHI agar medium supplemented with 0.5% glucose, 0.05% cysteine, and 1.5% agar, and grown in their respective culture conditions. The optical density (OD) of each dilution was read at a wavelength of 600 nm. The colonies formed on agar plates were quantified and associated with the OD_600_ measured for each dilution to prepare a standard curve. Prior to each experiment, bacterial concentrations were estimated using OD_600_. Planktonic culture of the five individual bacterial species was added to the apical chamber of the Transwell plates or in the 96-well plate at a concentration of 10^3^ CFU/mL in combination with the control or test conditions. Bacteria mix (BM) of the five species was prepared at a ratio of 30% *L. rhamnosus*, 30% *B. bifidum*, 20% *S. salivarius*, 15% *E. faecalis,* and 5% *E. coli*. Each bacterial species was diluted to 10^3^ CFU/mL before combining in the said ratio to form the BM. The BM was combined with the control or test condition before experiments.

### 2.3. Food Model Preparation

The food model (FM) was based on the previously developed food matrix described by Zhang et al. [18]. The FM consists of the nutrients from a standard US diet. Sodium caseinate (Sigma, 3.44% *w*/*w*), a protein source, was dissolved in 10 mL of sterile 10 mM phosphate buffer solution, pH 7. The solution was homogenized using a homogenizer at medium speed for 45 min. The solution was substituted with 3.42% (*w*/*w*) corn oil (fat source) and another 10 mL of sterile 10 mM phosphate buffer solution, followed by homogenization at high speed for 15 min to obtain a fine emulsion. Finally, 0.7% (*w*/*w*) pectin (dietary fiber source), 5.15% (*w*/*w*) starch, 4.57% (*w*/*w*) sucrose (sugar source), and 0.534% sodium chloride (mineral source) were added to the emulsion with continuous stirring in between. The prepared nutrient emulsion was frozen at −80 °C overnight and freeze-dried using a freeze dryer for a week. The freeze-dried product was referred to as the food model and stored at −20 °C until needed. The FM was diluted and streaked on BHI agar plates to check for sterility. For experiments, 1 g of food model was subjected to in vitro digestion and added to the seeded assay plates.

### 2.4. Nanoparticle Preparation

E171 food-grade titanium dioxide nanoparticles (TiO_2_ NPs) were purchased from Fiori Colori (Aromata Group). The TiO_2_ NPs have been previously characterized [19]. TiO_2_ NPs were sonicated at a concentration of 1 mg/mL (prepared using sterile 18 MΩ water) for 2 min at 10% amplitude on ice. The sonicated solution was diluted 100 times in sterile 18 MΩ water, and 5.92 µL of the diluted NP solution was used for experiments to achieve a final dose of 1.25 × 10^−6^ mg/cm^2^ or 1.25 µg/cm^2^. The calculations are based on the surface area of the small intestine and average human intake of TiO_2_ [5,20]. TiO_2_ was combined with 1 g of food model to represent the FM + TiO_2_ condition.

### 2.5. In Vitro Digestion

The control and test conditions were digested in vitro in a sterile environment prior to experimental exposure. A serum-free, 5 mM glucose DMEM solution served as a control condition, while test conditions included FM, TiO_2_, and FM + TiO_2_. The in vitro digestion protocol has been previously described by Glahn et al. [21]. The weighed samples were dissolved in 10 mL of sterile 140 mM NaCl + 5 mL KCl with a pH of 2. The pH was then re-adjusted to 2 with sterile 0.1 M HCl. Next, 0.5 mL of a porcine pepsin solution (25 mg/mL) prepared in 0.1 M HCl was added to each sample. The samples were then incubated for 1 h at 37 °C, 5% CO_2_ on a rocker (55 oscillations/min). This constituted the “gastric” phase of digestion. This was followed by “intestinal” digestion, where the pH of the samples was increased to 5.5–6 using sterilized 0.1 M NaHCO_3_ or 1 M NaHCO_3_. A pancreatin bile solution was prepared using 2 mg/mL pancreatin (a mixture of trypsin, amylase, lipase, ribonuclease, and protease activities) and 11 mg/mL bile extract (a mixture of glycine and taurine conjugates of hyodeoxycholic and other bile acids) and filter sterilized with a 0.45 μm filter. Next, 2.5 mL of the prepared pancreatin-bile solution was added to each sample, and the pH was brought up to 6.9–7 using sterile 0.1 M NaHCO_3_ or 1 M NaHCO_3_. The volume of each sample tube was made up to 15 mL by weight using a sterile 140 mM NaCl + 5 mM KCl solution, pH 6.7. The samples were then referred to as digests and used for exposure studies.

### 2.6. Permeability Assay

The permeability assay was used to quantify the changes in barrier integrity using Lucifer yellow (LY, Life Technologies L453), a fluorescent paracellular permeability marker. LY stock of 1 mg/mL was prepared aseptically in 18 MΩ water and stored at 4 °C until needed. A standard curve of LY was plotted using 0, 10, 20, 30, 40, and 50 μM LY concentrations and the fluorescence values for each concentration. A 100 μL of digested sample was added to the apical chamber of Transwell plates with 100 μL of 50 μM LY. Samples (100 μL) were then collected from the basolateral chamber every 15 min for the first hour and every 30 min for the next three hours and transferred into a 96-well opaque black bottom plate. The removed 100 μL volume, when sampling, was substituted by adding 100 μL DMEM to the basolateral chamber of the Transwell plate. At the end of the exposure experiment, the 96-well plate was read at 485/560 nm using a Synergy 2 plate reader, controlled by Biotek’s Gen5™ Reader Control and Data Analysis Software (V1.11.5). The resulting fluorescence values of the samples and the standards were used to calculate the permeability (cm/s) across the intestinal monolayer.

### 2.7. Intestinal Alkaline Phosphatase (IAP) and Bradford Assays

IAP activity was determined using IAP and Bradford assay. The sample digests were added to the seeded 96-well plates and allowed a 4 h exposure at 37 °C, 5% CO_2_. At the end of the 4 h exposure to digested control or test conditions, the wells were washed with 1× PBS, followed by the addition of 200 μL of 18 MΩ water. The plate was then subjected to sonication at 4 °C for 15 min. The samples were scraped off the bottom of the wells and subsequently used for the assay. A standard curve was prepared with dilutions of a standard solution of 0.25 mg/mL p-nitrophenol, prepared in p-nitrophenyl phosphate (pNPP). The assay was carried out by adding 25 μL of sample or standard solution in 85 μL of pNPP solution to a 96-well clear bottom plate, which was then incubated for 1 h at room temperature. For the Bradford assay, a standard curve was prepared using a 1 mg/mL bovine serum albumin (BSA, Rockland, NY, USA) solution prepared in 18 MΩ water. The experimental setup included a 5 μL sample or standard solution and 250 μL of Bradford reagent (Sigma) in a 96-well clear bottom plate. The plate was incubated at room temperature for 15–30 min in the dark. The IAP assay and Bradford assay plates were read at 405 nm and 595 nm, respectively, using the Synergy 2 plate reader, controlled by Biotek’s Gen5™ Reader Control and Data Analysis Software.

### 2.8. Nutrient Transport Assay

The Caco-2 and HT29-MTX E12 cell monolayer was grown on Transwell membranes for 14 days before performing nutrient transport assays. During the 14-day period, tight junction integrity is established, the Caco-2 cells mature into enterocytes, and a mucus layer is formed by HT29-MTX E12 cells [22]. On day 15, the monolayers were exposed to digested FM and/or TiO_2_ samples or control solutions, in combination with 10^3^ CFU/well bacteria for 4 h. The DMEM medium used for the assays was serum free with 5 mM glucose. This ensured that the resulting protein and glucose content detected after the assays was a result of the nutrients in the food model. After the exposure period, samples were collected from the basolateral chamber for nutrient transport assays.

#### 2.8.1. Glucose Assay

The basolateral samples were subjected to charcoal treatment due to low glucose levels [23]. Activated charcoal (2% *w*/*w*) was added to the samples and mixed on a vortex for 30 s. The samples were then centrifuged for 30 s, and the supernatant was used for the assay. Prepared samples were added to glucose hexokinase liquid stable reagent (Infinity^TM^) at a ratio of 1:150 in a 24-well clear bottom plate. Samples were incubated for 3 min at 37 °C, and the absorbance was read at 340 nm using a Synergy 2 plate reader, controlled by Biotek’s Gen5™ Reader Control and Data Analysis Software. The absorbance values along with the glucose standard (1 mg/mL in 0.1% benzoic acid) were used to calculate the amount of glucose (μg) per mL of each sample.

#### 2.8.2. Protein Assay

The basolateral samples were subjected to charcoal treatment, as mentioned in Section 2.8.1. The protein content was assessed using a Bradford reagent. A 5 μL supernatant sample was added to a 96-well clear bottom plate with 250 μL of Bradford reagent. The protein assay was carried out as described in Section 2.6.

#### 2.8.3. Triglyceride Assay

The fatty acid content in the basolateral samples was detected using a triglyceride assay kit (Sigma, MAK266). A triglyceride standard curve was prepared using dilutions of a 0.2 mM standard solution. A 50 μL sample or standard solution was added to a 96-well clear bottom plate, along with a 2 μL lipase solution. Samples were incubated for 20 min at room temperature. A master mix was prepared using 46 μL TG assay buffer, 2 μL TG probe, and 2 μL TG enzyme mix per well. A 50 μL of the prepared master mix was added to the plate at the end of incubation. Samples were then incubated for 30–60 min at room temperature in the dark. The absorbance values were read at 570 nm using a Synergy 2 plate reader, controlled by Biotek’s Gen5™ Reader Control and Data Analysis Software. Using the standard curve, the triglyceride content (μg) per μL sample was calculated.

### 2.9. Bacterial Viability Quantification

The 4 h exposure to independent bacterial species and digested samples was followed by bacterial quantification. Bacterial viability was quantified using viable counts and quantitative PCR.

#### 2.9.1. Viable Counts

A 100 μL sample was collected from the apical and basolateral chambers of the Transwell plates. The samples were serially diluted with sterile 0.9% saline solution and plated on BHI agar plates using the drop-plate technique. The bacteria that adhered to the Caco-2/HT29-MTX E12 monolayer were harvested from the Transwell inserts. The wells were washed with 1× PBS, followed by cutting of the Transwell membrane. The membrane was then transferred to a 400 μL sterile saline solution and homogenized for 30 s per sample. The homogenized samples were diluted and plated on BHI agar plates. To allow selective growth of each bacterial species in the bacteria mix, the samples were plated on MRS agar for *L. rhamnosus*, MacConkey agar for *E. coli*, ESM agar for *E. faecalis*, using the drop plate technique, blood agar for *S. salivarius*, and BSM agar for *B. bifidum* using the spread plate technique. Plates were incubated at their respective optimum growth conditions for 24–48 h, based on the bacterial species, and bacterial colonies were quantified to determine the bacterial viability (total log CFU).

#### 2.9.2. qPCR

Samples for qPCR bacterial quantification were harvested by scraping into sterile saline, and cells were collected by centrifugation (1 min at 18,000× *g*,), the supernatant discarded, and the cell pellet stored at −80 °C. Genomic DNA (gDNA) was extracted according to manufacturer instructions (ZymoBIOMICS DNA Miniprep Kit), and samples were stored at −80 °C. qPCR was performed with the PerfeCTa^®^ SYBR^®^ Green FastMix^®^ (Quantabio), the primers for each bacterial species, and an Eppendorf Mastercycler Ep Realplex instrument. Primers were used to quantify *B. bifidum* (16s rRNA, FW ACGCGAAGAACCTTACCTGG, RV 5′-ATCTCACGACACGAGCTGAC-3′), *E. faecalis* (16s rRNA, FW 5′-TACGGCGAACATACAAAGCG-3′, RV 5′-ATTTTGAACACGCGACTGG-3′), *L. rhamnosus* (16s rRNA, FW 5′-AGACACGGCCCAAACTCTAC-3′, RV 5′-CGTTGCTCCATCAGACTTGC-3′), *S. salivarius* (Hypothetical Protein, FW 5′-TGCCTGGAACTAACGGTAGC-3′, RV 5′-CAACGACTTCAGAGCCTCCC-3′), and *E. coli* (*tnaA*, FW 5′-GACTGGACCATCGAGCAGAT-3′, RV 5′-CACGCAAAGGGTTCTGCACTC-3′). Standard curves were previously generated by relating CT values to viable cell counts and allowing experimental CT values to be interpolated into equivalent cell counts. In brief, cultures of target species were grown to stationary phase and quantified by drop plating. Genomic DNA was extracted from 1 mL of stationary-phase cultures and purified. DNA was serially diluted to produce standards of decreasing DNA concentration. Quantitative PCR was then performed on these standards, and the resulting CT values were plotted against viable cell counts. Standard curves were calculated to interpolate cell counts from experimental CT values.

### 2.10. Statistical Analysis

The data sets were tested for normality using the Shapiro–Wilk normality test and transformed using the Box–Cox transformation with appropriate conditions to achieve a normal distribution. The LY permeability results were analyzed using a one-way ANOVA. The IAP assay, bacterial viability, and nutrient transport assay results were compared using a two-way ANOVA followed by false discovery rate (FDR) correction using GraphPad Prism 8.3.1. FDR correction was performed using the Benjamini and Hochberg procedure [24] implemented in the R [25] package multi-test [26].

## 3. Results

### 3.1. Bacteria, Food Model, and TiO_2_ Affect Cell Permeability

Caco-2 and HT29-MTX E12 cells were seeded onto Transwell membranes to form an epithelial monolayer. The intestinal barrier was subjected to four different conditions: digested control (serum-free, 5 mM glucose DMEM, DM), freeze-dried food model (FM), physiologically relevant dose of TiO_2_ nanoparticles, and FM + TiO_2_ conditions. Permeability across the intestinal barrier was determined under these four conditions prior to the introduction of bacteria into the system to ensure that the synthetic food model was not negatively affecting the integrity of the barrier. The results show no significant changes in permeability compared to the control condition. In comparison to the food model, addition of TiO_2_ to FM showed a significant decrease in permeability (Figure 1A), while *L. rhamnosus* (Figure 1B), *E. coli* (Figure 1C), *S. salivarius* (Figure 1D), and *E. faecalis* (Figure 1F) did not cause any significant changes in permeability across the four conditions compared to the control. *B. bifidum* led to a significant decrease in LY permeability across all four conditions, indicating an improved barrier function (Figure 1E).

The five different bacteria were cultured independently and then combined in a physiologically relevant ratio to create a bacterial mock community (BM). The bacterial mix did not result in any significant changes in permeability (Figure 1G).

### 3.2. Bacteria Cause a Reduction in IAP Activity

The intestinal alkaline phosphatase activity was quantified after 4 h exposure. Compared to the control (serum-free, 5 mM glucose DMEM), the IAP activity following exposure to TiO_2_ and/or FM was not affected. The introduction of *L. rhamnosus* (Figure 2A) and *E. coli* (Figure 2B) reduced the IAP levels compared to their respective no-bacteria conditions. *S. salivarius* was the only bacteria of the five species studied that led to an increase in IAP activity compared to the no-bacteria conditions (Figure 2C). *B. bifidum* (Figure 2D) and *E. faecalis* (Figure 2E) both caused a decrease in IAP activity with the control, TiO_2_, and FM + TiO_2_ conditions compared to their respective no-bacteria conditions. In addition, *B. bifidum* with TiO_2_ NPs led to a reduction in IAP activity compared to the *B. bifidum* + FM condition. The bacterial mock community maintained the IAP activity compared to the no-bacteria condition, except with TiO_2_, which showed a decrease with the BM but was restored to the control levels by the addition of the food model (Figure 2F).

### 3.3. Bacterial Community Increases the Glucose Transport

The amount of glucose transported across the in vitro epithelial barrier was quantified after a 4 h exposure using a colorimetric glucose assessment assay. The control condition (serum-free, 5 mM glucose DMEM), FM, TiO_2_, and FM + TiO_2_ conditions displayed the same amount of glucose transported across the barrier. As shown in Figure 3A–C, the addition of *L. rhamnosus*, *E. coli,* or *S. salivarius* did not impact glucose transport. However, *B. bifidum* significantly reduced glucose transport in the FM + TiO_2_ condition (Figure 3D). In the presence of *E. faecalis,* on the other hand, the amount of glucose transported reduced with TiO_2_ but increased with the FM (Figure 3E). This is expected due to the additional amount of glucose added to the model as part of the FM available to be transported. In a mock community, the bacterial species increased the glucose transported compared to the respective no-bacteria conditions, with a slight reduction upon the addition of FM and TiO_2_ (Figure 3F), suggesting a difference in impact due to interspecies bacterial interaction.

### 3.4. Protein Transport Remains Unchanged with NP and Bacteria

Protein transport estimation conducted using the Bradford assay revealed no significant changes in protein transport across the Caco-2/HT29-MTX E12 monolayer in response to the FM, TiO_2_, or a combination of both. The inclusion of *L. rhamnosus* or *E. coli* did not impact protein transport (Figure 4A,B). *S. salivarius*, on the other hand, caused a slight increase in the amount of protein transported in FM + TiO_2_, but without statistical significance (Figure 4C). *B. bifidum* showed the opposite effect by decreasing protein transport without statistical significance, as depicted in Figure 4D. The inclusion of *E. faecalis* did not change the transport of protein compared to their respective no-bacteria conditions (Figure 4E). Upon combining the bacteria into a multi-species community, their exposure resulted in a decrease in protein transport, but without statistical significance (Figure 4F). The overall lack of change in protein transport indicates the disinvolvement of NPs or bacteria in protein transport in this model.

### 3.5. Bacteria Reverse the Effect of TiO_2_ on Triglyceride Transport

The last functional aspect of the epithelial barrier studied was the transport of triglycerides (TG), as fatty acids were a part of the synthetic food model prepared in this study. The conditions with FM showed a higher TG transport compared to the fatty acid content added to the study model as part of the FM, which contributes to the negligent level of TG transport seen in conditions without the FM. The significance is depicted to showcase differences between the control, FM, TiO_2_, and FM + TiO_2_ conditions for no-bacteria or bacteria conditions. Figure 5A shows a significant increase in the amount of TG with FM + TiO_2_ compared to the FM condition, which was reduced in the presence of *L. rhamnosus*. *E. coli* displayed the same result: a decrease in TG transport in the FM + TiO_2_ condition when compared to the no-bacteria condition. In addition, the addition of TiO_2_ to FM reduced the transport in combination with *E. coli* (Figure 5B). Compared to the no-bacteria condition, *S. salivarius* (Figure 5C), *B. bifidum* (Figure 5D), and *E. faecalis* (Figure 5E) reduced the TG transport in combination with the FM + TiO_2_ condition. Within the *B. bifidum* condition, TiO_2_ caused a decrease in TG transport with FM. Similar to the effect of independent bacterial species, the bacteria mix also led to a decrease in TG transport compared to the no-bacteria conditions with FM and FM + TiO_2_ (Figure 5F).

### 3.6. Bacteria Adhere to the In Vitro Intestinal Monolayer

With the assessment of the integrity and function of the epithelial barrier, it became pertinent to analyze the viability of the bacterial species added to the model. The viability tests were conducted on the apical, insert, and basolateral compartments of the Transwell plates. The results revealed the number of bacterial colonies in suspension that adhered to the monolayer and those that passed across the barrier, respectively. The initial concentration introduced into the model was 10^3^ CFU/well (or 3 log CFU). The viability assessment for *L. rhamnosus*, shown in Figure 6A, reveals no significant effect of the FM or TiO_2_ on the apical chamber population; however, the *L. rhamnosus* colony growth on the insert seemed to decrease in the presence of the FM. *E. coli* showed a relatively higher growth rate compared to *L. rhamnosus*. While the FM or TiO_2_ conditions did not impact *E. coli* growth in the apical chamber, its attachment to the insert was significantly reduced with TiO_2_ but not FM (Figure 6B). *E. coli* colonies were also seen to pass into the basolateral chamber, with the highest being in combination with FM + TiO_2_. As opposed to *L. rhamnosus* and *E. coli*, *S. salivarius* growth was scarce, pertaining to its strict conditions needed for optimum growth. *S. salivarius* colony growth in the apical and on the insert was favored in the absence of the FM, while only the control condition allowed the movement of the bacteria to the basolateral chamber (Figure 6C). *B. bifidum* growth, as shown in Figure 6D, was observed only on the inserts, with no significant difference between the four conditions studied. Lastly, *E. faecalis* survived and multiplied in the apical chamber and on the insert. But without any statistical difference between different conditions (Figure 6E).

### 3.7. Bacterial Survival as a Mock Community

The survival of each of the five bacterial species was assessed using PCR based population quantification (Figure 7). The results revealed a low viability of *S. salivarius* in the three chambers (apical, basolateral, and insert) of the Transwell, which can be attributed to its requirement for low oxygen growth conditions. *B. bifidum*, which did not survive well as a single species, was present in all three chambers. There was no significant effect of the food model or TiO_2_ on the viability of bacteria in the apical chamber. The attachment of *E. faecalis* was seen to slightly increase with TiO_2_, but without statistical significance. However, the presence of *E. faecalis* in the basolateral chamber increased significantly with FM + TiO_2_ compared to the control condition (serum-free, 5 mM glucose DMEM). The attachment of all bacterial species on the insert shows their interaction with the mucus layer and intestinal cells and suggests their involvement in the changes seen in barrier integrity and function. The overall CFU count for bacterial survival decreased when the bacterial species was present as part of the mock community compared to an independent bacterial culture, which could be due to the limited availability of nutrients for the bacteria community.

## 4. Discussion

Food-grade TiO_2_ was approved by the Food and Drug Administration (FDA) in 1966 as INS171 and by the European Union in 2006 as E171. However, there have been growing health concerns with the consumption of TiO_2_ NPs due to their potential impact on the intestinal tract, the gut microbiota, and their potential to cross the gut and cause an immune reaction in the body. The host GI tract provides a constant nutrient supply and stable growth environment to the microorganisms while gaining protection from the pathogens due to their competitive exclusion by the commensal bacteria, thus forming a symbiotic relationship [27].

The negative effects of TiO_2_ NPs on the GI are widely published [11,14,28]; however, the ingestion of such nanoparticles as part of a diet and their interaction with the gut microbiota have yet to be fully established. The properties of the NPs can change when added to food [29] and after being absorbed by the GI, resulting in varied effects [30,31]. In this study, a normal US diet was created in the lab using all the essential nutrients, as previously described by Zhang et al. [18]. This semi-synthetic meal, referred to as the food model (FM), represented a US diet with an additional intake of products rich in TiO_2_ NPs (E171 grade) at a realistic dose of 1.25 μg/cm^2^, calculated based on the average human intake of TiO_2_ NPs and the surface area of the small intestine [20,32]. The food model and TiO_2_ were subjected to a two-step in vitro digestion to create a physiologically relevant diet, which was then introduced into a Caco-2/HT29-MTX E12 intestinal model. Prior to exposure to the intestinal monolayer, the sterility of the food model was confirmed using the drop-plate culture technique. The food model or TiO_2_ did not have any significant impact on the permeability of the barrier compared to the control condition, which showed a permeability of 10^−5^ cm/s. However, when TiO_2_ was incorporated into the food model, the permeability decreased by 29% compared to the food model alone (Figure 1A).

The upper small intestine is predominantly composed of Gram-positive aerobes, and the terminal ileum has an equal number of aerobes and anaerobes such as *Bifidum* bacilli [2]. The Gram-positive lactobacilli comprise the majority of the normal flora in the jejunum. Lactobacilli, along with staphylococci and streptococci, are also present in the intestine distally [33]. *L. rhamnosus* [34,35] and *Bifidobacterium* spp. [36] are well-known probiotic species with positive effects on intestinal health. *E. faecalis* is a common commensal bacterium and is one of the first colonizers in an infant [37], while *E. coli* is a Gram-negative bacterium more prevalent in the colon and can also be found in high quantities in the small intestine [3]. The bacteria mix in this study was designed to include a variety of bacterial species ranging from aerobes (*Enterococcus* and *Escherichia*) to facultative anaerobes (*Staphylococcus, Lactobacillus*) and anaerobes (*Bifidobacterium*). The broad selection of bacterial species enabled the investigation of the host–diet–microbiota interaction as close to the in vivo scenario as feasible. In this study, the presence of *B. bifidum* resulted in a significant decrease in permeability across the in vitro intestinal barrier (Figure 1E), while the other bacteria did not impact the permeability, indicating a quick response of this probiotic species towards regulating intestinal health. IAP is expressed by the enterocytes primarily in the duodenum and is heavily involved in the regulation of gut homeostasis [38]. IAP can potentially restore the commensal gut bacteria through its ability to control the pH. IAP is a crucial contributor to the maintenance of gut homeostasis and health through interactions with the intestinal microbiota and diet. While the benefits of probiotic bacterial species such as *L. rhamnosus* and *B. bifidum* are widely established, the positive influence of other commensal species such as *E. coli* is not as well known. The positive impact of *E. coli* on IAP is suggestive of such benefits. The interaction between IAP levels and *E. coli* has also been published in an in vivo model wherein wild-type mice showed higher levels of *Lactobacillaceae* and commensal *E. coli* populations compared to IAP knock-out mice. Upon oral administration of IAP, the colonization of *E. coli* increased in the IAP knock-out population [39]. The food model and TiO_2_ did not cause any changes in endogenous IAP activity compared to the control condition. However, as previously reported [32], IAP was significantly reduced with the addition of *L. rhamnosus*, *E. coli*, *B. bifidum,* and *E. faecalis* (Figure 2), indicative of the changing gut environment with the addition of bacteria. The five bacterial species used in this study were combined to form a bacterial mock community (BM) at a physiologically relevant ratio of the bacterial population [40,41]. The co-culture of mammalian and bacterial cells was allowed for a short duration (4 h), during which the experiments were carried out. The BM did not significantly alter the human cell monolayer permeability (Figure 1G). As opposed to the change in IAP due to individual bacterial species, the bacterial mock community maintained the IAP at the same levels as no bacteria control in most cases (Figure 2F), suggesting that as a community, the bacteria did not negatively impact the gut environment.

The human small intestine, as the primary site of nutrient absorption, can lead to nutritional deficiencies and malnutrition due to any disruption in the intestinal barrier. Therefore, factors that influence intestinal health may also impact the body’s nutritional availability. The intestinal microbiome plays a significant role in the de novo biosynthesis and bioavailability of several nutrients [42]. In this study, three nutrients were selected to determine the relationship between NPs, diet, and gut microbiota. Glucose transport across the barrier was significantly increased with BM, while protein transport did not show any changes across conditions. Triglycerides do not traverse the cellular barrier due to their hydrophobicity. Instead, they are absorbed through hydrolysis into fatty acids and monoacylglycerol in the intestinal lumen. Intestinal epithelial cells, primarily in the upper villi [43], take up the lipids and resynthesize them to triglycerides, which are secreted out in the form of chylomicrons. One of the components of the food model was fatty acids, which cause higher triglyceride transport in the presence of the FM, with a further increase with TiO_2_ in the food matrix. Previous studies show a decrease in fatty acid absorption due to TiO_2_ NPs [19,44]; in contrast, our results show that when NPs are part of a food matrix, their effect on TG transport changes. This suggests that while TiO_2_ on its own may have a more adverse effect on intestinal permeability in this model, when consumed as part of a regular diet at a controlled level, it does not pose the same risk. However, further studies are needed to study not only TiO_2_ absorption but also its accumulation in the body to assess the overall risk factors. The change in transport of each of these nutrients was different in response to the different bacteria. As part of a common community, the bacteria showed positive or no significant changes in protein and TG transport, except for the FM + TiO_2_ condition. Glucose transport was enhanced with the bacterial mock community compared to the single bacteria condition (Figure 3). Microbe–microbe interactions can have combined effects on intestinal health. Literature suggests improved blood glucose levels in the presence of beneficial bacteria upon supplementation with pre- and probiotic cultures [45], which is seen in these results as well. Further supporting investigation with respect to insulin sensitivity and immune response would be needed, however, to determine this effect conclusively. Overall, the results confirm the successful incorporation of the bacterial mock community for acute exposure.

The duplication rate and the optimum growth conditions of bacteria differ from one another, but they still co-exist in the human gut. In this model, the individual bacterial populations do not cross the intestinal barrier, except for *E. coli* and *S. salivarius* (Figure 6); however, when in the bacterial mock community, all bacterial species were present in the basolateral chamber (Figure 7). Previously, a study published by Pinget et al. showed that TiO_2_ causes a decrease in colonic mucin 2 expression, which causes a reduced mucus layer leading to inflammatory reactions and gut dysbiosis [5,46]. This is also observed in the current study, where *E. coli* attachment on the insert, which can be interpreted as bacterial entrapment in the mucus layer in vivo, was reduced in the presence of TiO_2_ (Figure 6B). The acute exposure of the intestinal monolayer to both TiO_2_ and a bacterial community could be a contributing factor to the increased presence of bacteria across the barrier. *S. salivarius* is a facultative anaerobe whose presence was undetectable when cultured planktonically in the model due to its unique growth conditions. Since the Caco-2/HT29 experiments were conducted in a 5% CO_2_ incubator at 37 °C, the growth conditions were not optimal for *S. salivarius*. When cultured as part of the bacterial mock community, the *S. salivarius* species was able to survive, potentially due to microbe–microbe or microbe–mucosal interactions within the model [47]. *S. salivarius* migrated across the barrier when the monolayer was exposed to TiO_2_, but this occurrence was not observed when TiO_2_ was used as part of the food model, confirming the varying nature of NPs as a part of the food matrix. As a single species, *B. bifidum,* a strict anaerobe with a slow duplication rate, was detected only on the insert and not in the apical or basolateral chamber. Its presence on the inserts confirms that the effects on permeability and transport we see in this study are in fact due to its ability to attach to the monolayer. As part of the mock community, *B. bifidum* survival was enhanced, confirming the importance of using a bacterial mock community to get a more physiological representation. The attachment of all the bacterial species to the insert and their presence in the apical chamber confirm the successful incorporation of the bacterial community into the model.

## 5. Conclusions

The complex relationship between diet, intestinal health, and the microbiome is not fully understood or established. The impact of food additives is widely studied; however, each study consists of different experimental parameters, which prevents clear comparisons. In this work, we present a small intestinal model together with a standardized food model composed of all the components in a regular diet, which can be used as a base to study the independent contribution of dietary and environmental factors to gut health. As part of this food model, food-grade TiO_2_ did not cause a negative effect on the gut permeability but led to changes in bacterial entrapment and passage across the monolayer, indicative of its involvement in gut dysbiosis. We also showed the successful incorporation of five different bacterial species as a bacterial mock community, which maintained intestinal permeability and IAP levels and improved glucose transport across the barrier. An in vitro model with the capability to incorporate dietary and microbial interactions can be used not just for nutritional studies but also to build disease or malnourishment models with the possibility of studying how changes in microbial diversity counter those conditions.

## Figures and Tables

**Figure 1 microorganisms-11-01419-f001:**
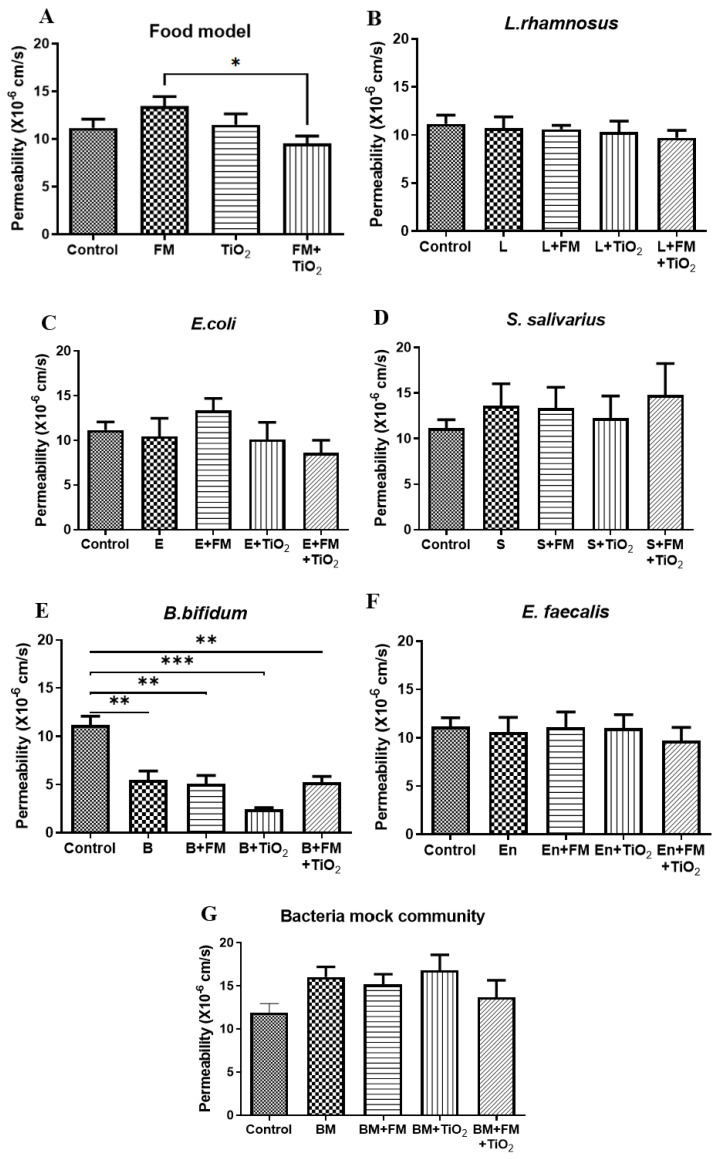
Permeability across the Caco-2/HT29-MTX E12 monolayer during 4 h exposure to (**A**) control (serum-free, 5 mM glucose DMEM), food model, TiO_2_, and food model + TiO_2_ in combination with (**B**) 10^3^ CFU/well *L. rhamnosus* (L), (**C**) 10^3^ CFU/well *E. coli* (E) (**D**) 10^3^ CFU/well *S. salivarius* (S) (**E**) 10^3^ CFU/well *B. bifidum* (B) (**F**) 10^3^ CFU/well *E. faecalis* (En) (**G**) 10^3^ CFU/well bacteria mix. Error bars represent SEM. FDR-adjusted *p*-values: * *p* < 0.05; ** *p* < 0.01; *** *p* < 0.001; using one-way ANOVA. n = 7. FM: food model, BM: bacteria mock community.

**Figure 2 microorganisms-11-01419-f002:**
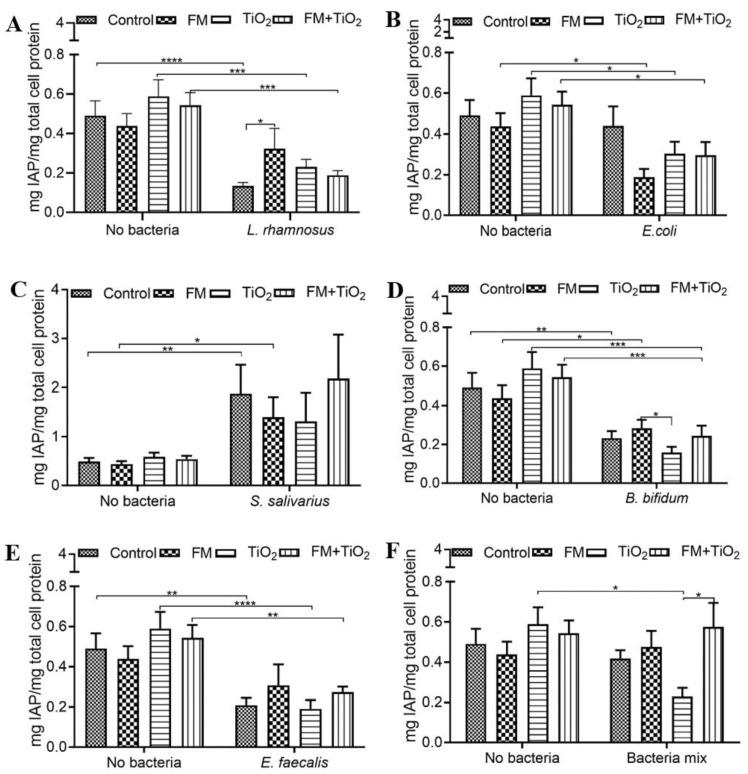
Intestinal alkaline phosphatase (IAP) activity following food matrix, bacteria, and/or TiO_2_ nanoparticle (NP) exposure. IAP activity after a 4 h exposure to control (serum-free, 5 mM glucose DMEM), food model, TiO_2_ nanoparticles, or food model + TiO_2_ with planktonic bacterial cultures (**A**) 10^3^ CFU/well *L. rhamnosus* (**B**) 10^3^ CFU/well *E. coli* (**C**) 10^3^ CFU/well *S. salivarius* (**D**) 10^3^ CFU/well *B. bifidum* (**E**) 10^3^ CFU/well *E. faecalis* (**F**) 10^3^ CFU/well bacteria mix. Error bars represent SEM. FDR-adjusted *p*-values: * *p* < 0.05; ** *p* < 0.01; *** *p* < 0.001; **** *p* < 0.0001 using two-way ANOVA with FDR correction IAP assay. n = 12. FM: food model.

**Figure 3 microorganisms-11-01419-f003:**
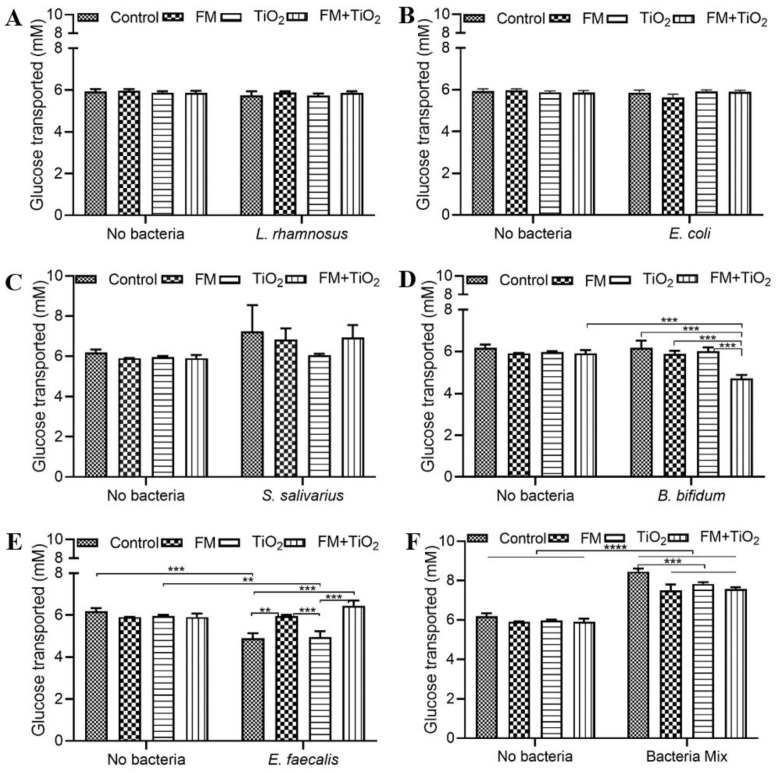
Glucose transport following food matrix, bacteria, and/or TiO_2_ nanoparticle (NP) exposure. The amount of glucose transported across the Caco-2/HT29-MTX E12 monolayer following a 4 h exposure to control (serum-free, 5 mM glucose DMEM), food model, TiO_2_ nanoparticles, or food model + TiO_2_ with planktonic bacterial culture of (**A**) 10^3^ CFU/well *L. rhamnosus* (**B**) 10^3^ CFU/well *E. coli* (**C**) 10^3^ CFU/well *S. salivarius* (**D**) 10^3^ CFU/well *B. bifidum* (**E**) 10^3^ CFU/well *E. faecalis* (**F**) 10^3^ CFU/well bacteria mix. Error bars represent SEM. FDR-adjusted *p*-values: ** *p* < 0.01; *** *p* < 0.001; **** *p* < 0.0001 using two-way ANOVA with FDR correction. n = 3. FM: food model.

**Figure 4 microorganisms-11-01419-f004:**
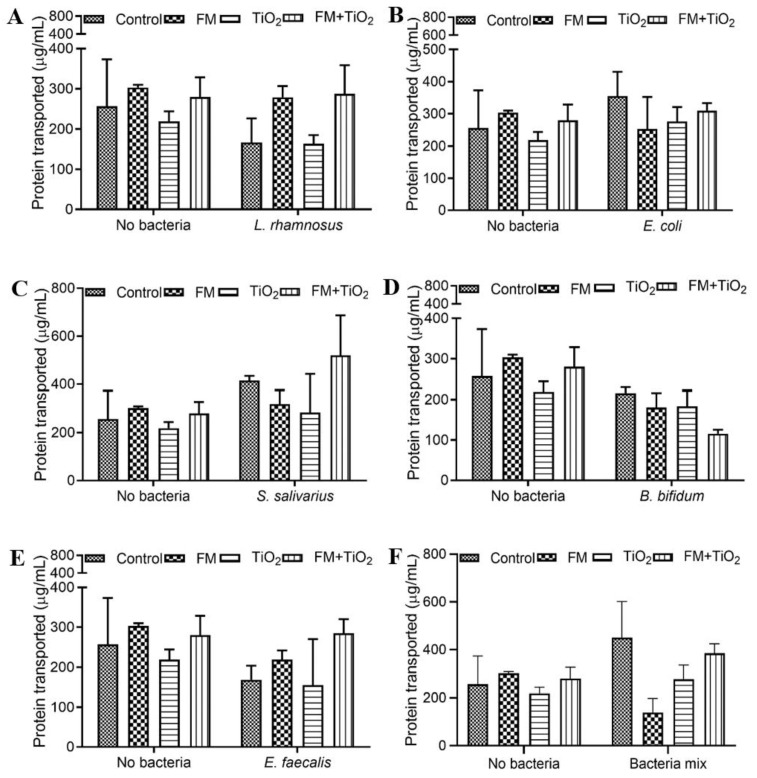
Protein transport following food matrix, bacteria, and/or TiO_2_ nanoparticle (NP) exposure. The amount of protein transported across the Caco-2/HT29-MTX E12 monolayer following a 4 h exposure to control (serum-free, 5 mM glucose DMEM), food model, TiO_2_ nanoparticles, and food model + TiO_2_ in combination with planktonic bacterial culture of (**A**) 10^3^ CFU/well *L. rhamnosus* (**B**) 10^3^ CFU/well *E. coli* (**C**) 10^3^ CFU/well *S. salivarius* (**D**) 10^3^ CFU/well *B. bifidum* (**E**) 10^3^ CFU/well *E. faecalis* (**F**) 10^3^ CFU/well bacteria mix. Error bars represent SEM. FDR-adjusted *p*-values using two-way ANOVA with FDR correction. n = 3. FM: food model.

**Figure 5 microorganisms-11-01419-f005:**
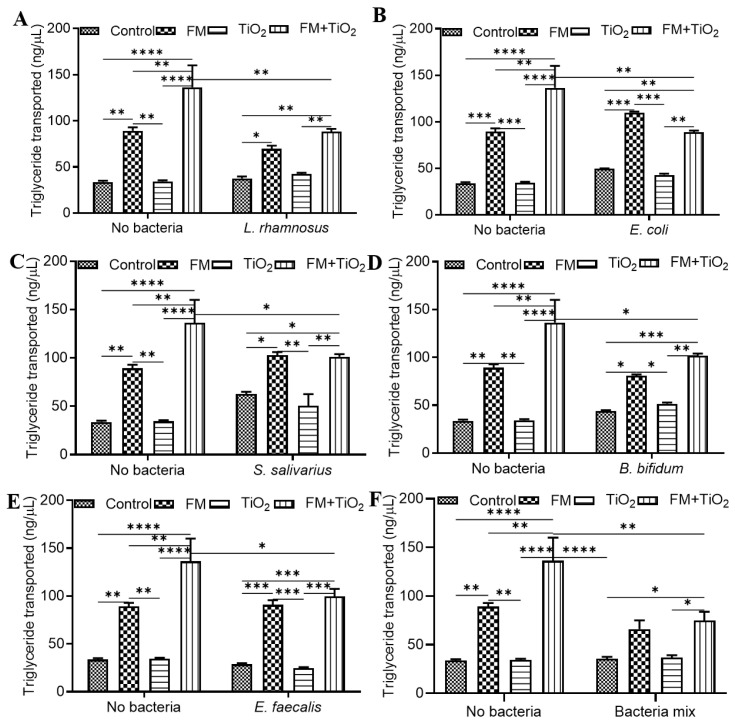
Triglyceride transport following food matrix, bacteria, and/or TiO_2_ nanoparticle (NP) exposure. The amount of triglyceride transported across the Caco-2/HT29-MTX E12 monolayer following a 4 h exposure to control (serum-free, 5 mM glucose DMEM), food model, TiO_2_ nanoparticles, or food model + TiO_2_ in combination with planktonic bacterial culture of (**A**) 10^3^ CFU/well *L. rhamnosus* (**B**) 10^3^ CFU/well *E. coli* (**C**) 10^3^ CFU/well *S. salivarius* (**D**) 10^3^ CFU/well *B. bifidum* (**E**) 10^3^ CFU/well *E. faecalis* (**F**) 10^3^ CFU/well bacteria mix. Error bars represent SEM. FDR-adjusted *p*-values: * *p* < 0.05; ** *p* < 0.01; *** *p* < 0.001; **** *p* < 0.0001 using two-way ANOVA with FDR correction. n = 3. FM: food model.

**Figure 6 microorganisms-11-01419-f006:**
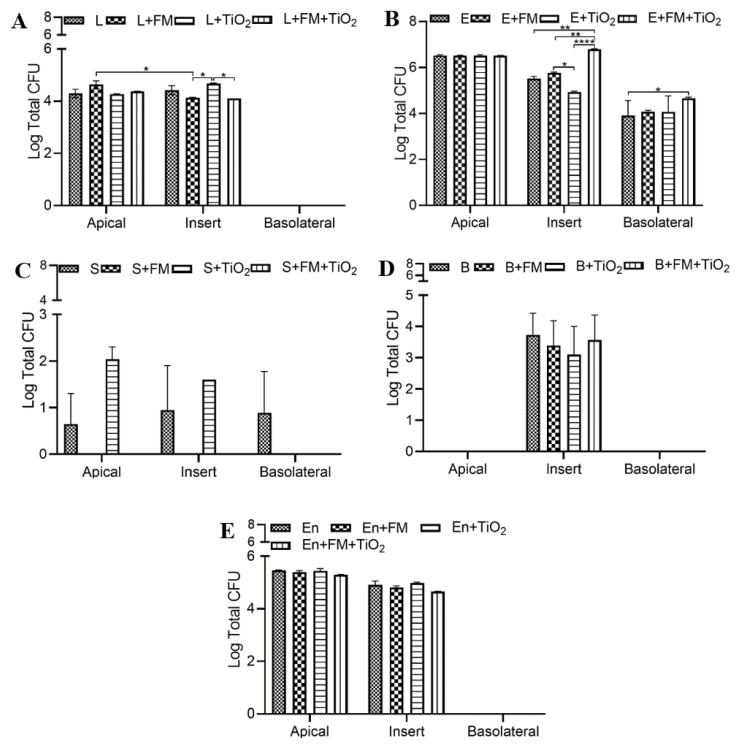
Bacterial viability quantification. The Caco-2/HT29-MTX E12 monolayer was exposed to control (serum-free, 5 mM glucose DMEM), food model, TiO_2_ nanoparticles, and food model + TiO_2_ for 4 h in combination with bacterial cultures. The bacterial viability was assessed in the apical, basolateral chamber, and insert of Transwell plates for planktonic bacterial culture of (**A**) 10^3^ CFU/well *L. rhamnosus* (L), (**B**) 10^3^ CFU/well *E. coli* (E), (**C**) 10^3^ CFU/well *S. salivarius* (S), (**D**) 10^3^ CFU/well *B. bifidum* (B), (**E**) 10^3^ CFU/well *E. faecalis* (En). Error bars represent SEM. FDR-adjusted *p*-values: * *p* < 0.05; ** *p* < 0.01; **** *p* < 0.0001 using two-way ANOVA with FDR correction. n = 3. FM: food model.

**Figure 7 microorganisms-11-01419-f007:**
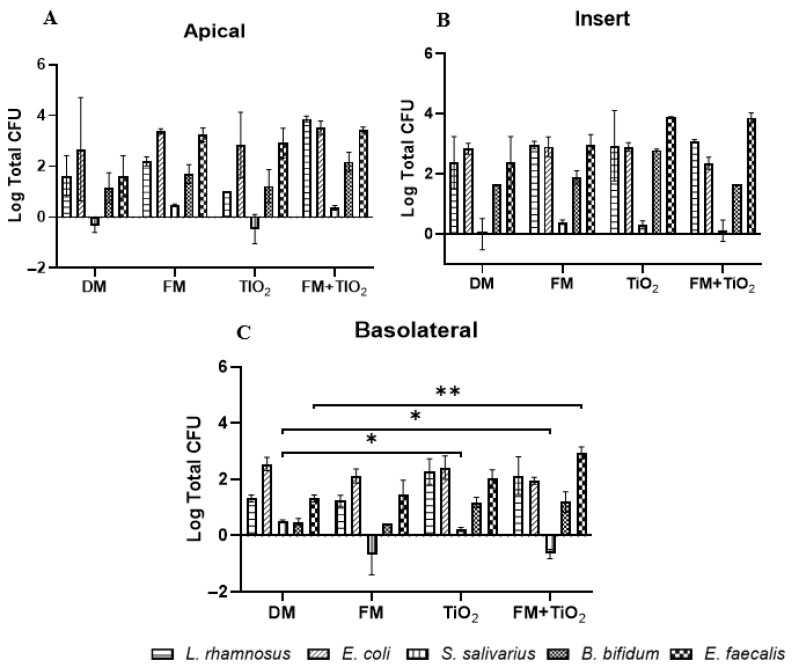
PCR-based bacterial quantification in a mock community. The Caco-2/HT29-MTX E12 monolayer was exposed to control (serum-free, 5 mM glucose DMEM), food model (FM), TiO_2_ nanoparticles, and food model + TiO_2_ for 4 h in combination with five different bacterial species as a bacteria mix with an initial concentration of 10^3^ CFU/well. The bacterial viability was assessed through PCR-based CFU count in the (**A**) apical (**B**) insert and (**C**) basolateral chamber. Error bars represent SEM. * *p* < 0.05; ** *p* < 0.01; using two-way ANOVA with FDR correction. n = 2.

## Data Availability

The data presented in this study are available on request from the corresponding author.

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
