# Peer review of "An In Vitro Small Intestine Model Incorporating a Food Matrix and Bacterial Mock Community for Intestinal Function Testing"

_microorganisms, 2023, doi:10.3390/microorganisms11061419_

Round 1
Reviewer 1 Report
In this manuscript, an in vitro Caco-2/HT29-MTX model was employed to investigate the interaction of food with the intestines and bacteria. The model examined the effects of titanium dioxide and bacteria on intestinal permeability, alkaline phosphatase, and nutrient transport. Below are specific areas where this manuscript needs to be improved:
1. It is necessary to add the reasons for selecting these five types of bacteria (Escherichia coli, Lacticaseibacillus rhamnosus, Streptococcus salivarius, Bifidobacterium bifidum and Enterococcus faecalis) as representatives in the paper.
2. Are all significant results shown in figure 1? Is there, for example, a significant change in figure 1A between the control group and the FM + TiO2 group?
3. In figure 2, the authors should explain why the result of Bifidobacterium bifidum as a probiotic was consistent with that of E. coli (resulting in decreased IAP activity).
4. In figure 3, mixed bacteria enhanced glucose transfer, in contrast to single bacteria. In the text (line 342), the author's description of this phenomena was overly simplistic, hoping to make more additions.
5. In figure 5, several couples were compared. The author used “*” to indicate their significance, which is messy. Please modify the presentation of the figure.
6. In line 482, it should be figure 1E, not 1D.
7. Why do individual bacterial populations, with the exception of E. coli, not traverse the intestinal barrier (Figure 6); nonetheless, all bacterial species were present in the basolateral chamber in the bacterial mock community (Figure 7).
Good
Author Response
We would like to thank the reviewer for taking the time to review our manuscript and for the helpful, constructive comments. We have taken all the comments into consideration and made corrections accordingly throughout the manuscript. The directed responses are outlined below:
- It is necessary to add the reasons for selecting these five types of bacteria (Escherichia coli, Lacticaseibacillus rhamnosus, Streptococcus salivarius, Bifidobacterium bifidum and Enterococcus faecalis) as representatives in the paper.
Response: To further support the rationale behind using the five bacterial species, additional reference and explanations have been added in line 474-476 (“The gram-positive lactobacilli…intestine distally”) and line 480-484 (“The bacteria mix in this study…as feasible”).
- Are all significant results shown in figure 1? Is there, for example, a significant change in figure 1A between the control group and the FM + TiO2group?
Response: The raw data was double checked by the author and yes, all the significant results have been reported in the figure.
- In figure 2, the authors should explain why the result of Bifidobacterium bifidum as a probiotic was consistent with that of E. coli (resulting in decreased IAP activity).
Response: Further explanation and supporting evidence have been added to the discussion in line 489-498 (“IAP can restore the … IAP knock-put population”) to relate the IAP trend observed with the different bacterial species.
- In figure 3, mixed bacteria enhanced glucose transfer, in contrast to single bacteria. In the text (line 342), the author's description of this phenomena was overly simplistic, hoping to make more additions.
Response: The comparison between single and mixed bacteria culture is further commented on in line 532-538 (“Glucose transport was … this effect conclusively”).
- In figure 5, several couples were compared. The author used “*” to indicate their significance, which is messy. Please modify the presentation of the figure.
Response: Thank you for the feedback. The concern has been addressed by increasing the font size of the asterisk and changing the graph to include only the relevant statistics. The statistics now show comparison of the effects of different diet conditions on triglyceride transport and the effect of bacterial presence in combination within each diet condition (e.g., the effect of FM+TiO2 in the no bacteria vs L. rhamnosus) instead of across diet conditions (e.g., the effect of FM+TiO2 in the no bacteria vs control in L. rhamnosus).
- In line 482, it should be figure 1E, not 1D.
Response: Thank you for pointing it out. The change has been made in the manuscript.
- Why do individual bacterial populations, with the exception of E. coli, not traverse the intestinal barrier (Figure 6); nonetheless, all bacterial species were present in the basolateral chamber in the bacterial mock community (Figure 7).
Response: E. coli is the only motile microorganism used in this study and as such, the one with a higher chance to cross the membrane if the integrity is lost. In addition, the bacterial presence as planktonic incorporation of single species into the gut model was determined using selective medium whereas the bacterial mock community was determined using PCR. This led to increased sensitivity and specificity in bacterial quantification. Furthermore, in the presence of the 5 bacterial species, E. coli could have provided the path for the other bacteria to cross the membrane. Independently of this, the E. coli presence in the basolateral chamber is high in both modes of detection.
Reviewer 2 Report
The paper An in vitro small intestine model incorporating a food matrix and bacterial mock community for intestinal function testing present a standardized dietary model composed of all components of a typical diet that can be used as a basis to study the independent contribution of dietary and environmental factors to gut health. This in vitro model with the ability to incorporate food and microbial interactions can be used not only for nutritional studies, but also to construct models of disease or malnutrition with the ability to alter microbial diversity to counteract these conditions.
Author Response
We would like to thank the reviewer for taking the time to review our manuscript and for approving it. We greatly appreciate it.